# Leaching of Phytochemicals from Beans during Hydration, Kinetics, and Modeling

**DOI:** 10.3390/foods13020354

**Published:** 2024-01-22

**Authors:** Gaurav Kumar, Dilini Perera, Kundukulangara Pulissery Sudheer, Pangzhen Zhang, Sushil Dhital

**Affiliations:** 1Bioresource Processing Research Institute of Australia (BioPRIA), Department of Chemical Engineering, Monash University, Clayton, VIC 3800, Australia; gaurav.kumar@monash.edu (G.K.); dilini.perera1@monash.edu (D.P.); 2Department of Agricultural Engineering, Kerala Agricultural University, Vellanikkara, Thrissur 680656, Kerala, India; kp.sudheer@kau.in; 3School of Agriculture & Food, Faculty of Veterinary & Agricultural Sciences, The University of Melbourne, Melbourne, VIC 3010, Australia; pangzhen.zhang@unimelb.edu.au

**Keywords:** wastewater, phenolic compounds, oligosaccharides, kinetic modeling, optimization

## Abstract

In the current era, there is a growing emphasis on the circular economy and the valorization of waste products. Bean processing industries generate substantial nutrient-rich waste laden with valuable phytochemicals. Understanding the leaching patterns and kinetics of major phytochemicals is key to designing better processes leading to increased sustainability. This review investigates phytochemical leaching mechanisms and kinetic modeling methods. Firstly we lay the foundation with a broad theoretical framework, and later deal with kinetic modeling approaches and promising areas for future research. Currently, the composition of industrial-scale bean wastewater remains undocumented in the open literature. Nonetheless, drawing from existing studies and general bean composition knowledge, we proposed a multi-phase leaching process. We hypothesize three distinct phases: initial leaching of phytochemicals from the outer seed coat, followed by a second phase involving polysaccharides, and concluding with a third phase wherein phenolic acids within the cotyledons leach into the hydration water. This review aims to shed light on the complex process of phytochemical leaching from common beans during hydration. By combining theoretical insights and practical modeling strategies, this work seeks to enhance our understanding of this phenomenon and ultimately contribute to the optimization of food processing methods with reduced environmental impact.

## 1. Introduction

Beans (*Phaseolus vulgaris*) belonging to the Fabaceae family are a sustainable and economic source of proteins, complex carbohydrates, vitamins, and minerals [1,2]. They have very long shelf life and can withstand slightly harsher storage conditions, making them a potent food source for securing nutritional security. Not only are beans highly nutritious, their cultivation also has a significantly lower environmental impact when compared to animal-based protein sources [3]. For instance, certain varieties of bean crops help in nitrogen fixation, contributing to the increased arability of the soils. These are also used during crop rotation cycles as they have the ability to fix nitrogen with higher drought tolerance and lower susceptibility to soil erosion [4,5,6]. Increasing the production and consumption of beans can help in addressing the sustainable development goals (SDG 2030), especially those relating to zero hunger (goal 2), good health and well-being (goal 3), and climate action (goal 13) [6].

Hydration is the first and the most important upstream processing operation during bean processing. It promotes the uniform distribution of heat during thermal operations like canning. During this process, beans are immersed in water for a variable duration ranging from several minutes to days. The primary aim is to facilitate the softening of the beans and eliminate anti-nutritional factors such as saponins, lectins, and phytates [7,8,9]. However, with these antinutritional factors, several nutritious compounds like polyphenolics, proteins, and fibers also leach out into the hydration water, which eventually ends up into the waste streams [9,10,11]. While several pressing reasons necessitate studying phytochemical leaching during hydration, we can class them into two categories: scientific reasoning and industrial reasoning. From the scientific reasoning point of view, studying waste streams can uncover the nutritional qualities, health implications, and environmental impacts. Moreover, it can pave the way to designing strategies for waste valorization. Industrial reasoning for studying waste water primarily stems from the cost efficiency and nutrient loss. Insights into waste water can help in designing and optimizing processing conditions to minimize nutrient loss and increase profitability margins. Likewise, they can help in uncovering the potential of aquafaba (bean wastewater) for culinary applications [12].

Various studies have identified the health-promoting effects of phytochemicals, especially phenolics and oligosaccharides; however, their leach out mechanism during hydration is not properly understood. The studies with an exclusive focus on the leached phytochemicals are very rare in the open literature. To address this gap in the present study, we have outlined the mechanism of phytochemical leach out and different leach-out kinetic modeling approaches. We have also identified numerous research gaps for fellow researchers to ponder upon.

## 2. Bean Hydration Process

When grains come in contact with water during hydration, water is taken up by the grains through a process known as “imbibition”. During imbibition, the water travels into the internal structures of the grain, leading to swelling and softening of the cell walls and tissues [9,13]. Grains are porous media consisting of a complex network of microscopic cracks and crevices, which can take the form of intercellular spaces or dents on the starch, proteins, or cell wall polysaccharide networks [11,14]. These pores are the major water uptake pathways during the imbibition process. Water molecules quickly fill the outer pores as a consequence of strong adhesive forces between the grain material and water. Almost simultaneously, as the water moves up the capillary column, the cohesive forces between water molecules pull up more water into the pores, triggering the imbibition process. When capillarity diminishes because water fills most pores, binary diffusion between the water and grain constituents starts, ultimately leading to the leaching out of various phytochemicals [9,15,16,17].

Beans, just like other leguminous seeds, have a distinctive seed structure with a protective outer layer, the seed coat. Seed coats have a hydrophobic cuticle layer, which hinders the swift entry of water molecules during the preliminary stage of imbibition [8,17,18]. The principal sites of water entry in beans are the micropyle and hilum due to their columnar structures, which lead the water into the interiors of the bean (Figure 1) [16,19]. Once the water enters the interior of beans, it travels to spaces between cotyledons and the cotyledon–seed coat interface. This results in quick hydration of the inner parenchyma cells of the seed coats which are much more susceptible to water uptake. This eventually hydrates the seed coat which undergoes glass transition and makes water penetration into the seed easier [15,17,20]. The hydration process is accompanied by the leaching out of phytochemicals, specifically phenolic compounds and oligosaccharides. It is also important to note that the specific compounds leaching out depend on the storage conditions of the beans and, most importantly, the temperature of hydration water [2,21]. For black beans [2], Giusti and co-workers found that the levels of certain phenolic compounds like (-)-catechin in the hydration water exceeded those found in the crude extracts from the beans. This might seem counterintuitive, but such observation indicates that larger compounds like procyanidins undergo hydrolytic degradation during hydration, resulting in the breakage of interflavan linkages (C4-C8), which produces (-)-catechin [22].

Studies also reported that when the beans were hydrated in room temperature, compounds like anthocyanins, some other flavonoids, tannins, and gallic acid leach out in excess; however, when the beans were hydrated in boiling water, the presence of other compounds like 4-hydroxy benzoic acid and syringic acids was identified. High temperatures may degrade and depolymerize large and complex phenolic compounds like lignin, which cause smaller phenolics (syringic acid), which otherwise may not, to leach out [2,23]. Some studies have identified that while hydration causes an outflux of phenolic compounds, mainly from the seed coat, it also induces an influx of the phenolic compounds, which result in an increased phenolic content of the cooked bean cotyledons [24,25].

Raffinose family oligosaccharides (RFOs) are the major class of oligosaccharides found in beans. Amongst these, stachyose, verbascose and raffinose are present in higher proportions. Earlier studies have associated the consumption of oligosaccharides with flatulence and stomach discomfort, largely due to the inability of intestinal enzymes to hydrolyze these molecules. In the human digestive tract, these molecules make their way to the large intestine. Here, they are fermented, causing the releases of gases like carbon dioxide, hydrogen, and methane, thus inducing flatulence [26]. Some recent studies have identified a rather beneficial effect of these oligosaccharides on human health. Researchers have identified that with the introduction of certain prebiotic bacterial strains like Bifidobacterium, large oligosaccharides can reach the human gut and be metabolized by the gut microbiome to produce short-chain fatty acids, which can positively affect gut health [27,28].

Perera and co-workers reported that hydration significantly increases the RFOs content in hydration water of two lupin varieties in the range of 20–80 times [11]. The study highlighted the dominant role of temperature in causing RFO to leach out during hydration. Increase in hydration temperature accelerates the mass transfer leading to a general increase in the phytochemical leaching out. Moreover, hydration triggers α-galactosidase activity, leading to enzymatic hydrolysis of the RFOs, and breaks them into smaller saccharide fractions like glucose and galactose and aids in phytochemical leach out. Prolonged hydration leads to germination, which can further reduce the RFOs content [29,30].

## 3. Hypothesized Mechanism of Phytochemical Leach out during Hydration

Phytochemical leach out is a prominent phenomenon observed during bean hydration. The spatial distributions of phytochemicals, especially polyphenolic compounds, are an important chemical attribute which is fundamental to uncovering the leach out mechanisms. While it is well known that seed coats are compositionally very different from cotyledons, the exact compounds present in these fractions greatly depends on the bean varieties. However, several studies show the presence of high concentrations of condensed tannins, anthocyanins, and flavanols including quercetins- and kaempferols-like compounds in the seed coats of various south-American bean varieties [31,32]. Similar studies on the seed coat of Australian adzuki beans revealed the presence of high amounts of flavonoid compounds like (-)-catechin, a prominent flavanol, and Rutin, a glycosylated flavonol [33]. Figure 2 and Table 1 shows some commonly occurring phytochemical compounds in common beans.

It is important to note that the seed coat compositions can also be influenced by the color of the beans. Troszyńska and co-workers found that the white peas did not contain condensed tannins and also had significantly different phenolic composition [34]. Rodriguez and co-workers seconded these observations and reported that flavonol compounds could not be detected in white-colored beans [35]. Cotyledons are rich in hydroxycinnamic acids, including ferulic acids, *p*-coumaric acids, gallic acids, and sinapic acids along with their derivatives, among others [32,35]. Based on this evidence, it could be hypothesized that phytochemical compounds like tannins, proanthocyanidins, quercetin, and kaempferol, which are primarily present in the seed coats, leach out during the hydration of the seed coat (Figure 3).

This may be called the first stage of phytochemical leach out. After the complete hydration of the seed coat, water starts filling into the microscopic cracks and voids present in the cotyledon and interacts with cell wall polysaccharides, leading to the hydrolytic degradation of these polysaccharides. This may start the second phase of leaching out, primarily raffinose family oligosaccharides, phytic acids, and lectins. Hydrolytic degradation of the cell wall materials and swelling, which results in high tortuosity, cause the excessive leach out of hydroxybenzoic and hydroxycinnamic phenolic acids as they are primarily located inside the cotyledons. This can be regarded as the third and final stage of phytochemical leach out. We would like to highlight the fact that these stages of leach out are hypothetical and have not been established. These stages are merely a theoretical framework for attaining a simple understanding of otherwise complex phenomena. Also, these stages are not mutually exclusive and may occur simultaneously, subject to the distribution profiles of these phytochemicals, which change with genetic variations and agronomic factors [35,36]. Analyzing the hydration water for phytochemicals at various time points during the hydration process can be undertaken to validate the proposed hypothesis. A prospective study can be conducted by sampling water at equally spaced time intervals, such as 15 min initially, with intervals extended to 1 h after 2–3 h subject to how quickly the bean hydrates. Subsequently, an untargeted screening of the water can be performed using liquid chromatography–tandem mass spectrometry (LC-MS/MS) to identify the present compounds. Upon identification, semi-quantification of these compounds can be achieved using high-performance liquid chromatography with diode array detection (HPLC-DAD) or high-performance liquid chromatography with refractive index detection (HPLC-RID), depending upon the compounds being quantified. This study can conclusively demonstrate patterns in which phytochemicals leach out.

**Table 1 foods-13-00354-t001:** Concise list of prominent and abundant phytochemicals present in common beans.

Common Phytochemicals	References
Syringic acid	[33]
*p*-Coumaric acid	
Ferulic acid	
Rutin	
Quercetin- derivatives	
Luteolin	
- (-) catechin	
kaempferol derivatives	[35]
Gallic acid	
delphinidin	
cyanidin	
petunidin	
pelargonidin	
malvidin aglycones	
Raffinose	[9,10,11]
Stachyose	
Verbascose	
Cellulose	
Hemi cellulose	
Pectin	
Lignin	
Tannins	

## 4. Modeling Approaches in Phytochemical Leach Out

Modeling is a cost-effective strategy which helps in various aspects of the design and optimization of food processing operations. Mathematical modeling can help in predicting the trend of leaching out of phenolics with respect to different processing conditions like temperature and treatment time and save lots of time and resources. In addition, modeling can be very helpful in attaining process uniformity and ensuring transferability [37,38]. There are numerous mathematical modeling approaches; two prominent ones that stand out in the case of phytochemical leach out are mechanistic modeling and phenomenological modeling approaches. Both approaches have their advantages and disadvantages and are used for different purposes. While the mechanistic model aims at understanding the mechanisms behind any phenomenon, the phenomenological model focuses on predicting an observed phenomenon without much attention to the underlying causes. The majority of studies dealing with the modeling of phytochemical leach out behavior from beans have undertaken phenomenological modeling. Therefore, we are suggesting a mechanistic framework for the leaching of phytochemicals during hydration, adapting from studies dealing with phenolic extraction from non-bean materials.

### 4.1. Mechanistic Modeling

As outlined in the previous section, the mechanistic modeling of leaching behavior makes clear assumptions concerning cause–effect relationships during the hydration process. Similar to extraction [39], leaching during hydration can be considered a two-step process, with a washing stage and a diffusion stage. The initial stage, where water is in contact with beans and starts to travel inside the bean structure to hydrate the inner parenchyma tissues, can be thought of as the washing stage. Once the beans are primed and hydrolytic degradation of cellular components begins to accelerate the dissolution of phytochemicals into water, it can be analogous to the second diffusion stage.

### 4.2. Major Assumptions in Existing Studies

Physical realities are very complex and pose major challenges when modeling any phenomenon. Thus, some assumptions have to be made in order to model these phenomena [37]. The major assumptions when modeling leach out kinetics include diffusion as the primary mode of mass transfer, unidirectional radial mass flow, uniform distribution of phytochemicals throughout the body beans, beans being perfectly spherical in shape, and negligible expansion of the beans during hydration. It is important to recognize that beans, as a natural biological material, exhibit anisotropic properties due to variations in tissue structures [40,41]. Additionally, the spatial distribution of water within beans can be quite complex and challenging to incorporate into modeling frameworks accurately. Consequently, a simplified approach may involve treating beans as isotropic media with a uniform distribution of initial moisture content. This simplification can make the modeling process more manageable, even though it does not capture the full complexity of the material.

### 4.3. Governing Equations

#### 4.3.1. Washing Stage

The washing stage, where the primary mode of mass transfer is dynamic convection, is generally considered fairly quick. The quantity of leached phenolics depends on contact surface area, temperature, and the initial phytochemical concentration in water [39,42,43]. The mass transport can be estimated using Equation (1):(1)$$\frac{dm_{P}}{dt}=k_{i}A\left(\omega_{s}-\omega_{w}\right),$$
where $m_{P}$ is the mass of extracted phytochemical (kg of phytochemical), $k_{i}$ is the coefficient of interaction between water and beans surfaces (kg of water m^−2^/s), A is the surface area of contact between water and beans (m^2^), $\omega_{s}$ dissolving coefficient at equilibrium (kg of phytochemical per kg of water), and $\omega_{w}$ is the concentration of phytochemicals actually dissolved in water (kg of phytochemical per kg of water).

The accessibility parameter $\delta X_{P}$ (kg of leached out compound/kg dry material) is the primary indicator of the washing stage and can be expressed as Equation (2).
(2)$$\delta X_{P}=k_{i}A\left(\omega_{s}-\omega_{w}\right)\delta t$$

As the name suggests, accessibility parameters measure how readily water can access the beans in the initial stage and start the leach out process. A higher $\delta X_{P}$ value suggests that initial phytochemicals are easily leaching out and vice versa. It is important to note that the value of $\delta X_{P}$ depends not only on the beans and water properties but also on the interactions between them and can change with any changes in these properties.

#### 4.3.2. Diffusion Stage

After the initial washing stage, diffusive mass transfer of phytochemicals starts. The generalized form of Fick’s second law of molecular diffusion can be represented by Equation (3):(3)$$\frac{\delta \rho_{p}}{\delta t}=D_{eff}\frac{\delta^{2}\rho_{p}}{\delta r^{2}},$$
where $\rho_{p}$ (kg/m^3^) is the apparent density of phytochemicals present in the beans, $D_{eff}$ (m^2^/s) is the effective diffusivity, t (s) is time, and r (m) is the radial coordinate. Term $\frac{\delta \rho_{p}}{\delta t}$ represents the rate of change of phytochemical density in beans, whereas $\frac{\delta^{2}\rho_{p}}{\delta r^{2}}$ shows the secondary derivative of the concentration profile with respect to the radial position, signifying the spatial dimension of how fast or slow is the leaching taking place. After solving these differential equations with [44] solutions, Equation (4) can be obtained:(4)$$\frac{M_{\infty }-M_{t}}{M_{\infty }-M_{i}}=\sum_{i=1}^{\infty }\frac{6}{i^{2}\pi^{2}}\;exp\left(\frac{-{i^{2}\pi }^{2}D_{eff}(t-t_{0})}{r^{2}}\right),$$
where $M_{\infty }$ (kg/kg db) is the quantity of maximum extractable phytochemicals, $M_{t}$ (kg/kg db) is the quantity at time *t*, $M_{i}$ ((kg/kg db) is the quantity of phytochemicals already present in hydration water before the commencement of the diffusion stage, and $t_{0}$ is the washing time. For ease of calculation, this infinite series can be limited to its first term, this leads to Equation (5).
(5)$$\frac{M_{\infty }-M_{t}}{M_{\infty }-M_{i}}=\frac{6}{\pi^{2}}\;exp\left(\frac{-\pi^{2}D_{eff}(t-t_{0})}{r^{2}}\right)$$

For further ease of calculation, we can linearize this equation by operating both the LHS and RHS with a natural log operator to obtain Equation (6).
(6)$$Ln\frac{M_{\infty }-M_{t}}{M_{\infty }-M_{i}}=Ln\frac{6}{\pi^{2}}-\left(\frac{\pi^{2}D_{eff}(t-t_{0})}{r^{2}}\right)$$

The experimental data can be plotted according to Equation (6), which will give the values of slope and intercepts. From the slope value, Dⅇff can be easily calculated from Equation (7).
(7)Deff  Slope ∗ r2π2

This framework can offer mechanistic insights into the leaching of phytochemicals. There are some inherent shortcomings of this approach; for instance, we have assumed that the sole mode of mass transfer is diffusion, there are limited geometrical considerations, and we are ignorant of bean swelling behavior during hydration. However, factoring in these considerations might make the calculations very complex, which might not be desirable. This approach should work very well and with high accuracy when dealing with bulk samples, as has been demonstrated in various other studies dealing with mass transfer phenomena of biomaterials.

## 5. Phenomenological Models

### 5.1. Power Law Extraction Model

This model is the most widely used model for the extraction of chemical compounds. It can be expressed as Equation (8) [45]:(8)$$C=kt^{n},$$
where *C* is the concentration of phytochemical species, *t* is the time, k is the proportionality constant, and *n* is the exponent factor to which *t* is raised and defines the nature of power law relationship. If *n* > 1, this indicates that an increase in t can cause a rapid increase in phytochemical concentration. If *n* < 1, the increase in concentration will be at a much slower rate. If the value of *n* = 1, the concentration has a linear relationship with time. The value *n* < 0 represents a negative relationship between phytochemical concentration vs. time, which is not possible in practical scenarios, considering that beans are hydrated in pure water with negligible initial phytochemical content.

### 5.2. Peleg’s Extraction Model

Peleg’s model is yet another widely used extraction model, which can find wide applicability to model leach out kinetics. It is a two-term model and can be expressed as Equation (9) [46,47,48]:(9)$$c=c_{0}+\frac{t}{k_{1}+k_{2}t},$$
where c is the concentration of phenolics at any time *t*, $c_{0}$ is the initial phenolic concentration, and $k_{1}$ and $k_{2}$ are the rate constants which signify the leach out rate and capacity constant, respectively. Since the initial concentration of phenolics $c_{0}$ in water is zero for most cases, we can rewrite Equation (9) as follows:(10)$$c=\frac{t}{k_{1}+k_{2}t}.$$

If the value of $k_{1}$ increases, this signifies the decrease in rate of mass transfer and vice versa. Similarly, an increase in $k_{1}$ indicates a decrease in the capacity of phenolics to leach out. The experimental concentration vs. time data can be fitted to this equation using a non-linear curve fit. Alternatively, Equation (10) can be linearized by taking the reciprocal on both the sides:(11)$$\frac{1}{c}=\frac{k_{1}+k_{2}t}{t},$$
which is of the form of y = mx + c as y = $\frac{1}{c}$, m = $k_{1}$, x = $\frac{1}{t}$, and c = $k_{2}$. The plot between $\frac{1}{c}$ and $\frac{1}{t}$ can give the values of $k_{1}$ and $k_{2}$ [49]. The major advantage of linear curve fitting is that it does not require an initial guess value, unlike non-linear models where an initial guess is necessary for obtaining iterative solutions. This can present some challenges, particularly when the range of constant values is unknown. However, we recommend the former approach as mass transfer is essentially a non-linear process. With the advent of various curve-fitting software tools, non-linear regression-based curve fitting has become more convenient and accurate. In a recent study on extraction of Mangiferin from *Mangifera indica* leaves, Anbalagan and co-workers reported that the Peleg’s extraction model was much better at predicting the extraction kinetics when compared to other conventional models like the logarithmic model and first- and second-order kinetic models [50]. Lin and co-workers also ascertained the suitability of Peleg’s model in predicting the extraction kinetics of major phenolics from white teas. However, the authors made an interesting observation that an increase in extraction temperature resulted in decreased goodness of fit for all the phenolics [51].

### 5.3. Weibull Extraction Model

The Weibull distribution model is a widely used statistical model, which can predict the leaching behavior with very high accuracy [45,52]. It is a non-linear model and is characterized by exponential decay or growth, which can be expressed as Equation (12):(12)$$C=\beta \left(1-e^{\left(-kt^{n}\right)}\right),$$
where β is the scaling factor and represents the magnitude of correlation between *C* and *t*; *k* is the rate parameter, which indicates the rate of mass transfer; and n is the shape parameters. If *n* < 1, the graph shows a downward concave behavior (sigmoidal) and represents a higher initial growth and plateaus towards the end; n = 1 means the curve follows an exponential behavior; and *n* > 1 shows an upward concave shape (parabolic) of the graph. The leach out kinetic graphs in most cases follow the sigmoidal behavior. Similar to other non-linear curve fitting procedures, the initial parameter estimation is required for this model as well. Lezcano and co-workers accessed the suitability of different kinetic models for phenolic leach out from Colombian propolis and concluded that Weibull extraction model was amongst the best-performing models. The authors observed that their distinct extraction phases were aptly captured by the Weibull model. The first phase had fast growth with a significantly higher slope, attributable to the large concentration gradient. The second phase was characterized by a slowly reducing slope, and the third and last phase had a stable trend due to the attainment of an equilibrium condition [53]. Similar observations were reported for modeling the phenolic leach out from onion wastes by Roman and co-workers. The Weibull model outperformed the power law model as it converges to factor in the equilibrium state attained towards the end of leach out process, while the power law does not approach a limit and keeps on increasing with respect to time [54].

### 5.4. Two-Site Kinetic Model

This model can be considered a quasi-phenomenological model, as this differs from other kinetics models in terms of its approach. This model can be expressed as Equation (13) [55]:(13)$$\frac{c}{c_{0}}=1-[Fe^{-k_{1}t}]-[\left(1-F\right)e^{-k_{2}t}],$$
where c and $c_{0}$ are the polyphenolic concentration at t and initial polyphenolic concentration, $k_{1}$ and $k_{2}$ are the lumped parameters of mass transfer rates, *F* is the fraction of phenolics leaching at a faster rate, and (1 − *F*) denotes the remaining fractions which leach out at a slower rate [56,57]. In this model, two-stages of phenolic leach out are considered, unlike Crank’s model, which considers a uniform distribution of phytochemicals. The first stage is represented by the first term, which is a first-order kinetic equation with the term *k*_1_. This stage mimics the phytochemical leach out from the broken or degraded cells. Since these compounds have high accessibility to water, this stage is characterized by a faster mass transfer rate. The second stage is represented by the last term, which captures the phytochemical leach out from the intact cells, therefore having a slower rate of mass transfer [55,58,59].

## 6. Future Scope

While numerous studies related to extraction kinetics of major phytochemicals from bio-products are available, no studies dealing with leach out kinetic modeling of common beans are reported. If hydration is carried out at a lower pH to reduce microbial load, there is a likelihood of cell wall degradation. Similarly, at higher temperatures, there is a possibility that enzymes may degrade during the hydration process. However, at lower temperatures, the reduced water availability slows down the process, thereby restricting enzyme activity. Individual compounds, chemical complexes, and the degree of hydrolysis of cell wall polysaccharides all represent crucial information which is not readily available in the open literature. Due to this, very limited information is available about the composition of these waste waters. Additionally, no definitive patterns in the way phytochemicals leach out during hydration has been identified. Due to these constraints, the development of a robust leach out model that explains its mechanistic underpinnings is extremely difficult. If we generate the primary data of these attributes, advanced modeling approaches including computational modeling can be undertaken. This is a major research gap and needs to be addressed by the wider scientific community. Likewise, more emphasis should be given to designing a multiple-stage hydration process to yield maximum benefit from the waste water. One such study could investigate the feasibility of a two-step hydration process, one for the recovery of phytochemicals and the other for polysaccharides. Similarly, follow-up studies on the cost benefit/ratio of an such approach and strategies by which to further improve can lead us one step closer to the United Nation’s sustainable development goals 2030.

With the impending climate change catastrophe, it is imperative that non-thermal processing alternatives of conventional energy and heat-intensive operation must be explored. There is a dearth of such studies, especially in the legume processing domain. Non-thermal processes have shown promising results when applied to other food processing operations, most notably the application of ultrasounds in extraction industries and the applications of pulsed electric field treatment in the potato processing industry. While there are numerous studies on the hydration kinetics and textural attributes of non-thermal technologies like ultrasound, their effects on phytochemical leaching out are largely underexplored. Similarly, the applications of other prominent non-thermal technologies like pulsed electric field processing, high pressure processing, and cold plasma processing are also limited. There is a need for such study in order to understand how different treatments affect the hydration behavior and the leach out behavior of phytochemicals so that their application in industrial applications can be undertaken.

Although phytochemical leaching out is a widely known phenomenon, no comprehensive technoeconomic study has been conducted to investigate the loss of phytochemicals. Such studies can help in developing better resource allocation strategies and also foster efficient resource utilization. Moreover, an estimate of the lost phytochemicals can also aid in process optimization and design and in the assessment of the environmental impact of the overall hydration process. Such studies can nudge the scientific community towards the development of separation strategies to isolate beneficial phytochemicals from the waste streams.

## 7. Concluding Remarks

Common beans are a rich and cost-effective source of proteins, complex carbohydrates, and other beneficial phytochemicals. Regular consumption of legume-rich diets can offer numerous health benefits. However, during hydration, a significant portion of these phytochemicals, especially phenolic compounds and dietary fibers, leach out from beans into the hydration water. So far, the composition of industrial-scale bean wastewater has not been reported in the open literature. However, based on some studies and our knowledge about the general composition of beans, we can suggest that the leaching out of phytochemicals can occur in multiple phases. We hypothesize that there may be three such leach out phases. In the first phase, phytochemicals present in the outer seed coat (condensed tannins, kaempferol, etc.) may leach, followed by polysaccharides in the second stage, and finally, the phenolic acids present in the cotyledons may leach out. To better understand the leaching mechanism, identification of compounds, along with an investigation into the leach out kinetics, is required. We recommend adopting the modeling approaches highlighted in this article as a starting point and anticipate the development of a much more advanced modeling framework over time. This can provide critical insights into these areas and lead to the development of more efficient processes with much lower environmental impact.

## Figures and Tables

**Figure 1 foods-13-00354-f001:**
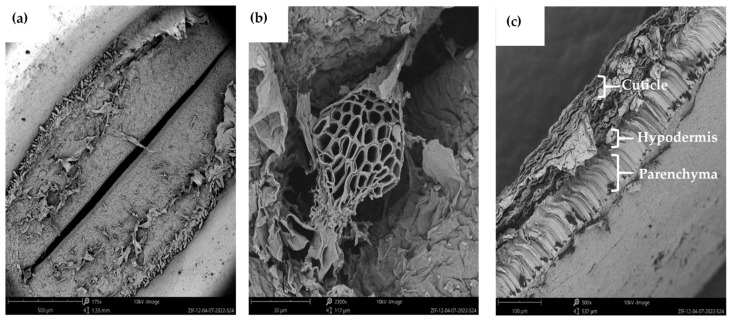
Scanning electron micrographs illustrating the (**a**) hilum, (**b**) micropyle, and (**c**) different tissue layers of the lupin bean seed coat (adapted from Perera and co-workers [11]).

**Figure 2 foods-13-00354-f002:**
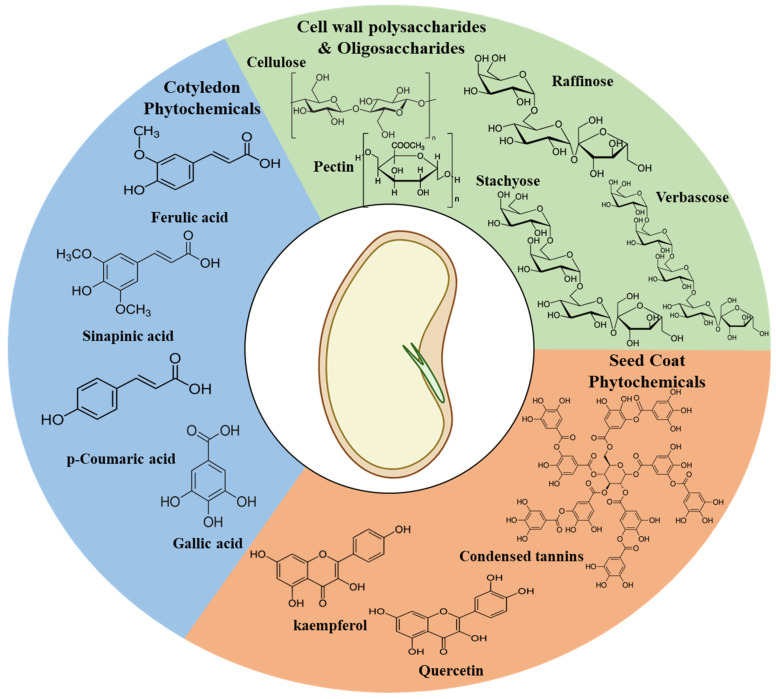
General spatial distribution of some prominent phytochemicals in beans.

**Figure 3 foods-13-00354-f003:**
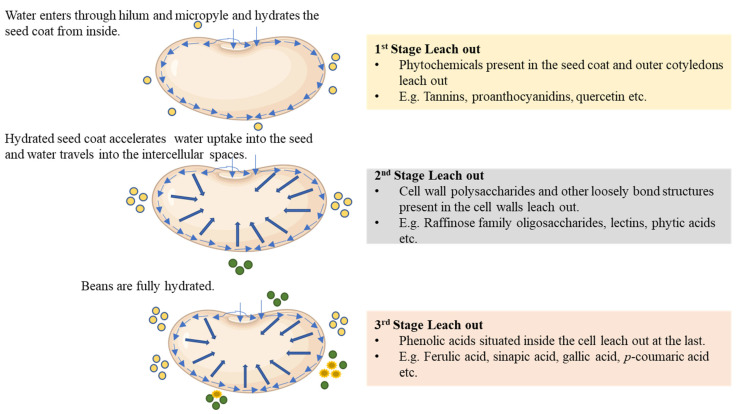
Hypothesized mechanism of phenolic leach out in different stages from hydrated beans.

## Data Availability

No new data were created or analyzed in this study. Data sharing is not applicable to this article.

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
