# Peer review of "Leaching of Phytochemicals from Beans during Hydration, Kinetics, and Modeling"

_foods, 2024, doi:10.3390/foods13020354_

Round 1

Reviewer 1 Report

Comments and Suggestions for Authors

The manuscript is clear and well presented and it gives insights on possible future research on beans phytochemicals and kinetics during hydration process. the modelling part needs further research and data. The authors should clarify more the importance of the proposed model and its applications. A table summarizing the known phytochemicals and corresponding references should be included. 

Reviewer 2 Report

Comments and Suggestions for Authors

The manuscript refers to leaching of compounds during the soaking of beans. It is a well written work that covers the basic biochemical aspects related to the compounds that diffuse and the mathematical models to explain/predict their diffusion. However, it present minor flaws.

Keywords: Please, avoid repeating the same words present in the tittle. To improve the manuscript visibility, replace "beans" and "hydration".

Introduction

line 45-47 The sentence is difficult to understand. please, divide into two sentences.

Bean hydration process

Line 83 It would improve the manuscript quality if the different tissues were represented in Figure

Line 101 - How much lignin do beans contain? What are the phenolic acids structurally constituting the beans lignin?

Hypothesised mechanism of phytochemical leach out during hydration.

Line137 - Please, standardize the nomenclature of phenolic compounds. (-)-catechin

Figure 1 - "p" in p-coumaric acid should be in italic. The section cell wall polysaccahrides contain also oligosaccharides. This should be revised to differentiate polysaccharides from oligosaccharides.

Line 147 p-coumaric acid 

Figure 2 2nd stage refers the hydrolysis of cell wall polysaccharides. at what extent are they hydrolysed? Don't they refer to structures that are less bond to cell walls and that are easily extracted with water? "p-coumaric acid"

Future/Conclusion:

Considering the published data, is it possible to identify the compounds more prone to diffuse? Not in the general term, but rather specific compounds. How can the hydration process be designed to yield a waste water of value? Can the hydration process be designed into 2 steps? One for the recovery of phytochemicals and other for polysaccharides? Is the cost benefit/ratio of such approach acceptable? How can it be improved? These are some aspects that the mansucript coud explore, enhancing its novelty and the generation of new knowledge.

References Please revise

3,4,9,10,12,14,17,20,21,26,30,33,34,38-42, 46-49, 51,52,55-59 not abbreviated journal name

Species names should also be presented in italic

Round 2

Reviewer 2 Report

Comments and Suggestions for Authors

The authors improved the manuscript and answered to the issues raised. I would only suggest to revise the reference to hydrolysis (lines 360-362). Instead, the authors should mention that acid hydrolysis might occur if using high temperatures and acidic conditions during soaking. Some cell wall degradation by enzymes might also occur, but at very limited extent. This is supported by the low water availability along the seed, that slowly increases during soaking, restricting the enzyme activity. 
